

# Optimizing forensic file classification: enhancing SFCS with $\beta_k$ hyperparameter tuning

D. Paul Joseph[1] and Viswanathan Perumal[2]

[1] School of Computer Science Engineering and Information Systems, Vellore Institute of Technology University, Vellore, Tamilnadu, India
[2] Department of IoT, School of Computer Science and Engineering, Vellore Institute of Technology University, Vellore, Tamilnadu, India

## ABSTRACT

In forensic topical modelling, the $\alpha$ parameter controls the distribution of topics in documents. However, low, high, or incorrect values of $\alpha$ lead to topic sparsity, model overfitting, and suboptimal topic distribution. To control the word distribution across topics, the $\beta$ parameter is introduced. However, low, high, or inappropriate $\beta$ values lead to sparse distribution, disjointed topics, and abundant highly probable words. The $\beta_j$ parameter, in conjunction with seed-guided words based on Term Frequency and Inverse Document Frequency, is introduced to address the issues. Nevertheless, the data often suffers from skewness or noise due to frequent co-occurrences of unrelated polysemic word pairs generated using Pointwise Mutual Information. By integrating $\alpha$, $\beta$, and $\beta_j$ into file classification systems, classification models converge to local optima with O(n log n* |V|) time complexity. To combat these challenges, this research proposes the SDOT Forensic Classification System (SFCS) with a functional parameter $\beta_k$ that identifies seed words by evaluating semantic and contextual similarity of word vectors. As a result, the topic distribution ($\Theta_d$) is compelled to model the curated seed words within the distribution, generating pertinent topics. Incorporating $\beta_k$ into SFCS allowed the proposed model to remove 278 k irrelevant files from the *corpus* and identify 5.6 k suspicious files by extracting 700 blacklisted keywords. Furthermore, this research implemented hyperparameter optimization and hyperplane maximization, resulting in a file classification accuracy of 94.6%, 94.4% precision and 96.8% recall within O(n log n) complexity.

# INTRODUCTION

Digital forensics (DF) uses scientific techniques and best practices to preserve, collect, validate, identify, analyse, interpret, document, and present digital evidence from various digital sources (*Kent, Chevalier & Grance, 2006*). The goal of DF is to provide accurate data that can be used to solve crimes, prosecute perpetrators, and deter similar incidents in the future. As DF is about collecting digital data, the quantity of data that requires collecting, analysing, and storing is increasing exponentially. Therefore, it dramatically compromises DF investigations' time, speed, and efficiency. To address this problem, *Rowe (2016)* coined the term 'uninteresting files' and suggested that identifying forensically relevant files

Corresponding author
Viswanathan Perumal, pviswanathan@vit.ac.in

reduces the time delay. In the forensic context, files that do not aid in investigations and those that do not contain personal or sensitive information are treated as 'uninteresting files'. To identify pertinent files and latent topics, topic modelling (TM) algorithms such as Latent Dirichlet Allocation (LDA), non-negative matrix factorisation (NMF), Hierarchical Dirichlet Process (HDP), and probabilistic latent semantic analysis (PLSA) play a vital role. One such widely used algorithm is LDA, which is integrated into many research domains such as text analysis, document classification (*Rüdiger et al., 2022*), malware analysis (*Upadhyay, Gharghasheh & Nakhodchi, 2022*), and email analysis (*Sun et al., 2021*) *etc.*

In TM algorithms, $\alpha$ hyperparameter is implemented in LDA to control each document's topic distribution ($\theta_d$). Dirichlet distribution is defined as $\theta \sim$ Dirichlet ($\alpha$), where $\alpha$ is considered a parameter and $\theta$ is topic proportion (*Blei, Ng & Jordan, 2003*). Traditional DF investigations suffer from numerous issues such as-increased false positive rate, the less efficient hit rate for keywords, the lack of possibility to recognise semantic structures, and the high computational effort resulting in unfolding the $\alpha$ hyperparameter to the DF domain. The $\alpha$ parameter relaxes the importance of traditional keyword search in terms of time, cost, and manual intervention besides identifying latent words in a document (*Noel & Peterson, 2014*). However, when $\theta$ is drawn from $\alpha$ alone, the same distribution is observed on $D = \{D_i, .., D_{i+n}\}$, where $i = \{1, 2, .., n\}$, so that $\theta_d$ is similar across D, and each D is assumed to have equal topic distribution. As a result, the $\alpha$ relies more on word occurrences, due to which the $\theta$ becomes consistent across documents. The lower the value of $\alpha$, the sparser $\theta$ becomes, leading to single topic domination, which is the significant drawback of using a single hyperparameter $\alpha$.

As a result, the $\beta$ parameter is implemented to control word-topic distribution $p(\frac{w}{t})$ by generating word proportions in each topic (T). Dirichlet distribution is defined as $\phi \sim$ Dirichlet ($\beta$) (*Blei, Ng & Jordan, 2003*; *Dieng, Ruiz & Blei, 2020*). For each topic (T), $\beta_k$ is a vector of vocabulary (V) length that contains probabilities for generating $w \in$ V. The word distribution vector $\phi_k$ for any topic (k) is represented by $\phi_k = \phi_{k1}, \phi_{k2}, ...\phi_{kV}$ and the probability density is given in the Eq. (1).

$$Dir(\phi_k|\beta) = \varepsilon_0(\beta) \prod (\phi_{kv}^{\beta-1}) \tag{1}$$

where $\varepsilon_0(\beta)$ is a constant, and the value is $\leq 1$. $\varepsilon_0(\beta)$ can be expanded as $\frac{1}{B(\beta)^V}$.

Taking the logarithm and derivation of $Dir(\phi_k|\beta)$, $\frac{\beta-1}{\phi_{kv}}$ results in 0. This suggests that the distribution becomes sparser when $\beta \leq 1$ or produces less coherent topics when $\beta \geq 1$. Various models such as O-LDA (*AlSumait, Barbará & Domeniconi, 2008*), H-LDA (*Wang & Blei, 2009*), and Parallel LDA (*Liu et al., 2011*) are implemented based on $\alpha, \beta$ parameters. Despite their integration, TMs in massive datasets still faces challenges, such as learning numerous coherent words due to the long-tailed distribution of languages. As a result, the $\beta_j$ parameter is implemented to incorporate prior knowledge into the topic modelling in addition to $\alpha, \beta$ parameters (*Jagarlamudi, Daumé & Udupa, 2012*).

The TM algorithm with $\alpha, \beta$ parameters inevitably discovers the latent topics in a *corpus* of text documents without prior information about the topics (*Blei, Ng & Jordan, 2003*; *Vayansky & Kumar, 2020*). However, in many cases, researchers or domain experts have

prior knowledge or guidance regarding topics of interest. The $\beta_j$ parameter enables the incorporation of this prior knowledge in the form of topic seed words or phrases, denoted by a seed set $S$, to guide the algorithm to the desired topics. These seed words or phrases can be provided by domain experts or extracted from external sources such as ontologies or dictionaries. $\beta_j$ in LDA works by initialising the distributions of topic words $P\left(\frac{w_n}{z_n}\right)$ and document topics $P\left(\frac{z_n}{\theta}\right)$ using the seed words or phrases. Then, these distributions are iteratively updated based on the terms in the documents, similar to traditional $\alpha, \beta$ in LDA, and the probability mass function is given in Eq. (2).

$$f(\theta; \alpha, \beta_j) = \frac{(\theta^{\alpha-1})(1-\theta)^{\beta_j-1}}{B(\alpha, \beta_j)} \tag{2}$$

where $\alpha, \beta$ are the hyperparameters controlling the $P\left(\frac{z_n}{\theta}\right)$ distribution and $\beta$ is the $P\left(\frac{w_n}{z_n}\right)$ distribution with $S$ priors. $B$ is the beta function obtained from the product of the sum of the parameters and the gamma function ($\gamma$) of each parameter (a, b) as shown in the Eq. (3).

$$\beta(a, b) = \frac{\gamma(a)\gamma(b)}{\gamma(a+b)} \text{ and } \gamma(a, b) = (a-1)!; (b-1)! \tag{3}$$

During the topic assignment process, the algorithm assigns more weight to the seed words or phrases in $\beta_j$ as per Eqs. (2) and (3), which helps guide the algorithm to the desired topics. However, when seed words heavily influence the model in $\beta_j$, overfitting occurs, and thus, generated topics are less representative of the *corpus* (C). Secondly, $\beta_j$ introduces a level of bias, which can compromise the model's interpretability and result in topic irrelevancy. Thirdly, seeds in $\beta_j$ are extracted based on Pointwise mutual information, resulting in noisy data. To overcome these significant issues, the authors propose the $\beta_k$ Secure Hash-Dynamic Operator Pattern (SDOT) Forensic File Classification System model that incorporates an additional functional parameter $\beta_k$. This parameter is defined according to the forensic domain and extracts seed words by observing semantic and contextual similarity. Thus, the SDOT Forensic File Classification System (SFCS-$\beta_k$) can be helpful in digital forensics to identify relevant topics in large amounts of unstructured textual data when integrated with the authors' previous forensic framework SDOT (*Joseph & Viswanathan, 2023*). Furthermore, this research answers the following questions.

1. Why do topical modeling algorithms prefer seed words during classification?
2. How do seed words affect the classification and accuracy of the model?
3. Should seed words be preserved to expedite forensic investigations?
4. How can we determine whether a file is relevant or irrelevant to a forensic investigation so that irrelevant files can be eliminated to conserve time while relevant files can be admitted as evidence in court?

This study presents the significant contributions to the above research questions.

1. Custom data pre-processing: This study developed a custom pre-processing method based on a forensic stop-words module to identify forensic relevant files in an RDC.

2. Forensic text analysis: This research integrated topical modelling and natural language processing algorithms to perform document analysis, document summarization, and identifying relevant forensic information in *corpus*.

3. Parametric Extraction module: This research proposed $\beta_k$ parameter in addition to $\alpha, \beta, \beta_j$ to identify competent seed-guided words concerning the forensic domain by integrating Word2Vec and fast cosine similarity. Seed words belonging to multiple topics are extracted to expedite the investigation process. This is the first work to integrate a seed-guided topical model into the digital forensics domain for file classification.

4. Effective labelling and classification: This research incorporated the $\beta_k$ parameter into the SDOT forensic file classification system to enhance file labelling, improve file classification and optmize the overall accuracy.

Forensic investigations consume considerable time due to the massive amount of data. To address this limitation, the authors introduce forensic seed parameter $\beta_k$ based on seed-guided model (*Jagarlamudi, Daumé & Udupa, 2012*) to automatically identify and extract forensic-relevant keywords. Initially, a ground truth dataset consisting of seed words belonging to eight categories defined by DHS is constructed and stored in the $\beta_k$ parameter. Then, using LDA with Word-to-Vector (W2V), the *corpus* is vectorized, and the words identical to seeds in $\beta_k$ are extracted based on Cosine similarity. Thus, all the identified relevant words are updated in $\beta_k$ according to their categories. Based on these seed words, the dataset is labelled and preliminarily classified with the help of the Keyword-Metadata-Pattern (KMP) classifier (*Joseph & Viswanathan, 2023*). Then, the authors used a linear support vector machine (L-SVM) to classify the forensic data into relevant or irrelevant files based on their nature. Furthermore, the proposed parameter $\beta_k$ is integrated into the author's previous work-SDOT system to achieve optimal results in classification, time complexity, and receding manual intervention. This study is the first to incorporate a guided topical model into the digital forensics domain to achieve optimal classification in less time, thus speeding up forensic investigations.

Therefore, this study aims to develop and evaluate the methodologies in digital forensics with a primary focus on forensic file identification, analysis and classification in disc forensics. The primary objectives of this research are:

1. To analyze the existing literature on digital forensics, focusing on forensic-relevant based filtering techniques.

2. To assess the efficacy of existing file classification techniques in digital forensics.

3. To examine the impact of the proposed file classification framework with the existing frameworks relevant to the DF domain.

The structure of the article is as follows: "Related Work" provides most related work with $\alpha, \beta, \beta_j$ parameters, while "SFCS-$\beta_k$ Forensic File Classification System" proposes the

seed parameter $\beta_k$ and how it works. "Experimental Results and Analysis" contains the experimental evaluation of the individual modules of the Parametric Extraction Module (PEM), the integrated three-stage overall system, and the results. In contrast, "Discussion" discusses the results and their interpretation, the system's performance and future work, and concludes with the Conclusion section.

## RELATED WORK

Data-less text filtering and classification (DFC) (*Li et al., 2018*) is implemented with $\beta_j$ parameter, assuming that each document is associated with a single category topic and multiple general topics. $\beta_j$ in DFC identifies highly relevant documents based on the seed parameter S. The seed parameter ($S \cong \beta_j$) with $S^L \cong \beta_j^L$ and $S^D \cong \beta_j^D$ is implemented in this work with two sets of seed words denoting the category label and description, where $\beta_j^D$ is manually composed with the domain knowledge. The seed words are extracted and used as the main source for document classification. In a *corpus*, documents $D$ with defined categories $C$, $\beta_{j_c}$ defines the set of small seed words, and the seeds are filtered, which fall outside $S_C$.

Let $\beta_j = U_r S_r$, where $\beta_j$ denotes the set of seed words, then the distance used to calculate between $\beta_j$ and the latent topic $k$ is defined in Eq. (4).

$$dist(k, \beta_j) = 1 - \sum_{s \in \beta_j} \sum_{w \in W_k} P_{LDA}\left(\frac{w}{k}\right) P_{LDA}\left(\frac{k}{s}\right). \tag{4}$$

The problem in the above equation is that $W_k$ is limited to 10 in DFC, which is the top 10 thematic words, and $dist(k, \beta_j)$ is relatively high for $W_k \geq 10$, where latent topics of irrelevant category $C$ are observed, that affects the optimised classification. Due to the constraint of the parameter $(W_k)$ in filtering noisy latent topics, the LDA model is forced to generate T over bounded controls. Another problem lies in $P_{LDA}(w|k)$ and $P_{LDA}(k|s)$ so that the total computational cost increases when $L \geq 3$ as mentioned in Eq. (5).

$$rel(w, c) = \frac{1}{\beta_{j_c}} \sum_{s \in \beta_{j_c}} \sum_{k \in L_s} P_{LDA}\left(\frac{w}{k}\right) P_{LDA}\left(\frac{k}{s}\right). \tag{5}$$

Furthermore, regardless of the size of $\beta_{j_c}$, rel(w, c) has no further influence on a higher classification rate if $\beta_{j_c} \notin D$ and $\beta_{j_c} \notin P_{LDA}(k|s)$ so that $D$ can only be classified as relevant category $C$ if the rel(w, c) score is high.

The probability for the category word and category topic is defined as $\delta_{w,c}$, where $\rho \in [0, 1]$ is given in Eq. (6).

$$\delta_{w,c} = \frac{\tau_{w,c}\rho}{1 - \rho + \tau_{w,c}\rho}. \tag{6}$$

Another con of the Eq. (6) is

$$\prod \begin{cases} \delta_{w,c} < 1, \text{DFC degrades} \\ \delta_{w,c} = 1, \text{DFC} \sim \phi^c \end{cases}. \tag{7}$$

This article overcomes the above problems by calculating semantical and contextual values using Word2Vec (W2V) and integrating them with the weighted vector $V_i$.

Seed Guided Topic Discovery (SEE-Topic) framework (*Zhang et al., 2022*) implemented $\beta_j$ with the functional parameter $(C \cong \beta_j)$ to generalise seed-guided topic discovery by allowing seeds outside the vocabulary. This framework leverages existing trained models and semantics from the input *corpus*. Given a *corpus* D, let $V_D$ be a set of terms in D with the underlying assumption that terms can be either words or phrases; the task of this work is to find out internal vocabulary terms $S_i \in V_D$ for each category $\beta_j^C = \{c_1, c_2, ..., c_{|\beta_j^C|}\}$.

$$\mathscr{T} = \sum_{d \in \mathscr{D}} \sum_{w_i \in d} \sum_{w_j \in \beta_j^C(w_i,h)} p(w_j \mid w_i) + \sum_{d \in \mathscr{D}} \sum_{w \in d} p(d \mid w') + \sum_{c_i \in \beta_j^C} \sum_{w \in S_i} p(c_i \mid w) \qquad (8)$$

The last part of the Eq. (8) is the proposed work that represents the similarity between the category $C_i$ and its representative term $S_i$. Performing a soft-maximization of the above equation, one obtains

$$U_w^T[V_w; V_d; V_c] = [X_{ww}; X_{wd}; X_{wc}] \qquad (9)$$

where the columns of $U_w, V_w, V_d, V_c$ are $u_{wi}, v_{wj}, v_d, v_{ci}$ respectively $(w_i, w_j \in V_D, d \in D, c_i \in \beta_j^C)$.

Since the numerator in the Eq. (10) denotes the vector multiplication of co-occurrence of $w_i$ and $w_j$ in D and the total terms in D, the resulting ensemble score parameter $\rho$ worsens as the value increases from 0.1 to 0.9

$$\mathbf{X}_{ww} = \left[ \log \left( \frac{\mathscr{D}(w_i, w_j) \cdot \lambda_{\mathscr{D}}}{\mathscr{D}(w_i) \cdot \mathscr{D}(w_j) \cdot b} \right) \right]_{w_i, w_j \in \mathscr{V}_{\mathscr{D}}} \qquad (10)$$

To address the above concerns, the authors propose multiple seeds set specific to the forensic domain $(\beta_k)$, besides tuning $\alpha, \beta,$ and $\beta_j$.

Tagged LDA (TLDA) (*Rani & Lobiyal, 2021*) implemented $\beta_j$ to summarise text using various sentence weighting techniques such as Lexical LDA for iterative retrieval of all possible topics; relative sentence weighting LDA for fetching topics within a single epoch and assigning weights; integrated sentence weighting LDA for integrating extracted latent topics; and sliding window-based sentence LDA, which operates on a sliding window protocol. TLDA is implemented by first calculating topic diversity of sentence $(\beta_j^{S_j})$ by $V_i = [p_{\{i,1\}}, f_{\{i,2\}}, ........, p_{\{i,k\}}]$ where $V_i$ is a vector with $k$ dimensions and $p_{i,j}$ is the frequency of occurrence of topic $T_j$ in sentence $\beta_j^{S_i}$. To find the similarity of latent topics, the authors computed the cosine similarity among vectors $\beta_j^{S_i}$ and $\beta_j^{S_{\{i,...,n\}}}$ by Eq. (11).

$$\cos(\beta_j^{S_i}, \beta_j^{S_{\{i+1\}}}) = \frac{<\beta_j^{S_i}, \beta_j^{S_{\{i+1\}}}>}{norm(\beta_j^{S_i}) * norm(\beta_j^{S_{\{i+1\}}})} \qquad (11)$$

where $norm(\beta_j^{S_i})$ is the Euclidean normalisation of $\beta_j^{S_i}$. The authors then calculated the final topic density score using Eq. (12).

$$TD_{score} = \frac{1}{n(n-1)} \sum_{i=1}^{n} \sum_{j=1, j \neq i}^{n} \cos(V_i, V_i). \tag{12}$$

One downside of this approach is that the parameter settings for LDA are unclear. Therefore, it is hard to apply the method to new datasets. Secondly, since it relies on support vector regression to rank sentences, overfitting is commonly seen during training when $\frac{1}{N_{train}} * \sum_i (Y_{train}[i] - Y_{ptrain}[i])^2$ is less than $\frac{1}{N_{test}} * \sum_i (Y_{test}[i] - Y_{ptest}[i])^2$. Thirdly, TLDA computes $P(A|B) = P(B|A) * \frac{P(A)}{P(B)}$ of latent variables $z$ over observed words $w$, which is formidable due to the high dimensionality of the joint space resulting $p^q$ combinations, where $q$ represents a variable that takes $p$ different values.

Pandemic LDA (P-LDA) (*Gupta & Katarya, 2021*) implemented $\beta_{jk}$ to capture the relationships between topics and the brands in the *corpus* that represents the relationship strength between brand (j) and topic (k). The parameter $\beta_{jb}$ is then incorporated into conditional word probability for each document (D) to capture multi-brand topics as shown in Eq. (13).

$$P(w|z, \beta) = \prod_{d=1}^{D_i} \prod_{n=1}^{W_n} \sum_{k=1}^{K} <p(z_{dn} = k), p(w_{dn}|z_{dn} = k, \beta)>. \tag{13}$$

P-LDA also used Dirichlet prior ($\beta$) over $\beta_{jb}$ to incorporate prior knowledge of brands. Even though the proposed parameter $\beta_{jb}$ captures complex relations between brands and topics from the reviews, this model performs poorly when the $K$ value is inappropriate. For example, if $0 < K_1 < K_2$, then

$$\prod \begin{cases} |T(K_1)| < |T(K_2)|, \text{case}:1 \\ |T(K_2)| > |D|, \text{case}:2 \end{cases}. \tag{14}$$

case:1 denotes that P-LDA poorly selects $D_r(D_r \sim$ relevant topics), whereas case: 2 denotes that P-LDA overfits $D$ and generates irrelevant $K$. Here, $D$ denotes data, and $T(K)$ indicates the topics generated by P-LDA with the hyperparameter $K$.

Another major limitation of this model is that it assumes that for each review, K = 1. Through our research, we found that some reviews cover multiple topics. Therefore, it is imperative to note that when K exceeds 1 for each review, the parameter $\beta_{bk}$ cannot capture the necessary relations, resulting in a significant increase in false negatives. Finally, this model results in $O(c^n)$ time complexity ($n$ represents variables and $c$ represents the highest cardinality) to construct joint probability distribution as shown in Eq. (15).

$$P(w, d, \theta, \phi|\alpha, \beta) = P(\theta|\alpha) \prod_{i=1}^{N} P(\mathbf{w_i}|\theta, \phi_{d_i}) P(\phi_{d_i}|\beta) P(d_i) \tag{15}$$

where *w* represents the collection of all reviews, *d* represents the collection of all brand descriptions, $\alpha$ and $\beta$ are hyper-parameters, $\theta$ is topic distribution, and $\phi$ is word distribution. The variable *N* is the total number of reviews in a *corpus*.

To overcome these concerns, the authors propose a $\beta_k$ parameter that assumes $K > 1$ for each *D*, thereby reducing false negatives.

## SFCS-$\beta_k$ FORENSIC FILE CLASSIFICATION SYSTEM

This article proposes a three-tier classification framework with functional seed parameter $\beta_k$ to automatically identify and extract seed parameters. In tier 1, the data is extracted and given to forensic triage. This triage is based on the author's previous work (*Joseph & Norman, 2019*) that performs disk triage, memory forensic triage, and multimedia forensics. However, the authors limit this work to disk forensic triage for file classification. Basic pre-processing is observed once the disk image is loaded, followed by forensic seed extraction. The seed extraction module extracts all the seed words trained with the W2V and the FCSS algorithm. Based on the results, the seed words are classified into eight categories defined by the Department of Homeland Security (DHS). Based on these seed sets, the *corpus T* is trained with the blacklisted keywords module, and the results are stored in a central repository. The functioning of the metadata and pattern modules can be found in *Joseph & Viswanathan (2023)*. The *corpus* is then passed to a support vector machine (SVM) based on the training for file classification. Thus, when integrated with the author's previous work, the proposed approach is represented as SFCS-$\beta_k$, enabling better file classification in forensic investigations. In this article, SFCS-$\beta_k$ describes the overall three-tier architecture, where the pre-processing and Parametric Extraction module constitute tier 1, the KMP classifier in tier 2, and the machine learning classifier training in tier 3. The overall architecture, represented as SFCS-$\beta_k$, is illustrated in Fig. 1 and is explained in the following sections.

### Pre-processing

To extract the $\beta_k$ parameter, the authors converted the *corpus* into a machine-readable format using a pre-processing engine, as shown in Fig. 1. $T_{OK}$ module known as Tokenization helps convert each word into an individual token. Case normalization is used in $T_{OK}$ to convert data into lower case, except for proper nouns, to eliminate the case sensitivity and make data consistent. $S_{TOP}$ is the stop words module, which removes all the common words in addition to conjunctions, prepositions, pronouns, *etc*. Custom-expanding contractions are developed in this work at phrase-level analysis to make data more readable and understandable.

$L_{emma}$, known as lemmatisation, is used in this work to extract the root words of a specific word without the meaning loss. The last module, $I_{ndex}$, is the Indexing module, which indexes all the extracted tokens into the central repository. In this work, the authors chose inverted indexing, which has substantially faster retrieval and storage times than other indexing strategies.
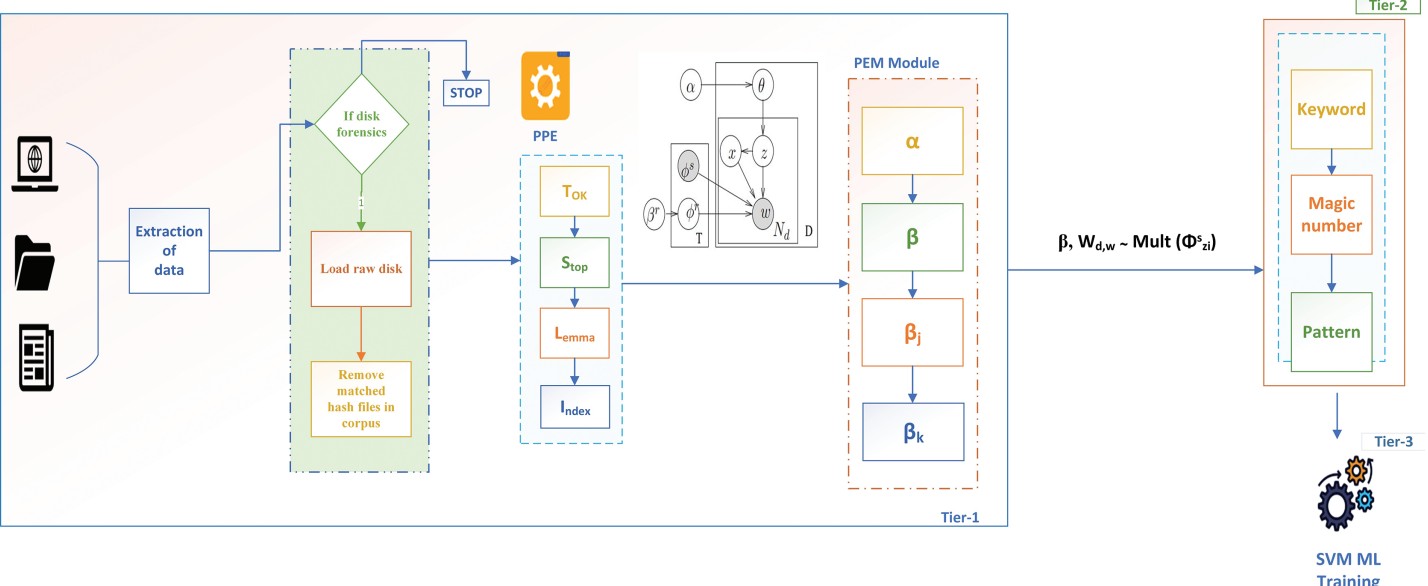

**Figure 1** Three-tier SFCS-$\beta_k$ architecture with $\alpha$, $\beta$, $\beta_j$, and $\beta_k$ hyperparameters.

## Parametric extraction module

Existing methods or traditional guided algorithms for topical modelling have enlisted the help of experts in extracting seed words manually. There have been few guided models for automatically extracting computationally intensive seed words. The guided LDA algorithm works by initialising the $P\left(\frac{w_n}{z_n}\right)$ distribution and the $P\left(\frac{z_n}{\theta}\right)$ distribution using the seed words or phrases. Seed words are the relevant words to each topic fed to the LDA model so that the model assigns weights and extracts relevant topics efficiently. The inclusion of priors or seeds greatly improves the quality of topic discovery. In such cases, it is difficult for the model to successfully identify the irrelevant topics if they are not supported by the appropriate seed words, resulting in poorer classification performance. It is also known that the quality of seed words plays a crucial role in the classification of documents and that using more seed words does not lead to a good classification rate (*Li et al., 2018*). Therefore, to overcome the problems with seed word extraction quality and increased time complexity, the authors proposed another functional parameter $\beta_k$ that controls the $P\left(\frac{w_n}{z_n}\right)$ distribution. The proposed $\beta_k$ parameter is explained as follows.

1. In standard LDA (*Blei, Ng & Jordan, 2003*), each document $D$ is represented as a mixture of $K$ topics, where $K$ denotes the total number of topics.
2. For each $d$, the mixture of topics is represented by $\theta_d$, which is a K-dimensional probability.
3. Each $w$ (word) in $d$ is generated by selecting topic $z$ from $\theta_d$, followed by selecting a $w$ from the vocabulary distribution of topic $\phi_z$, which is technically represented as

$$\theta_d \sim Dirichlet(\alpha),$$

$$z_{dn} \sim Multinomial(\theta_d),$$
$$w_{dn} \sim Multinomial(\phi_{z_{dn}}).$$

4. The overall sparsity of $\theta_d$ is controlled by the hyperparameter $\alpha$, which results in numerous problems, as discussed in previous sections.

Therefore, the authors introduced the $\beta_k$ hyperparameter to extract the seed words from the *corpus* (C) to overcome the issues. The plate notation of the proposed work is shown in Fig. 2, which depicts the proposed parameter $\beta_k$ in the guided LDA. It can be observed that $\beta$ contains two parameters known as $\beta_j$ and $\beta_k$, which represent a set of seed words that guide the entire *corpus*.

### Role and tuning of hyper parameters

1) $\alpha$–this parameter is used for document-topic density, typically set between 0.1 and 1.0, based on the number of topics. A smaller $\alpha$ (0.1) leads to a sparse distribution of topics within the documents (fewer topics per document), whereas a larger $\alpha$(1.0) encourages a more even distribution of topics across documents (more topics per document). In these experiments, the $\alpha$ value is assigned to 0.7 as each document is believed to hold multiple topics.

2) $\beta$–this parameter is used for topic word density typically set between 0.01 and 0.1, depending on the vocabulary size. A smaller $\beta_j$ leads to a sparser model where topics are associated with fewer words. A larger $\beta_j$ implies that topics will utilize a broader range of words. In these experiments, the $\beta_j$ value is set to 0.09 as the topics are known to have distinct vocabulary.

3) $\beta_j, \beta_k$–this parameter is used to store the domain-specific seed-guided words and their probabilistic distribution across the topics. Initially, the $\beta_j$ parameter can be initiated with the pre-defined or case-specific topics with higher probabilities to favour the words significantly and the remaining topics being assigned with low probabilities. This parameter can be integrated with topic-word distribution ($\beta$) as a weighted combination as given in the Eq. (16).

$$\beta_k = (1 - \lambda_s).\beta + \lambda_s * \beta_j \tag{16}$$

where $\lambda_s$ is a hyperparameter that controls the seed word influence within the range of $\lambda_s \in \{0, 1\}$. Initiating $\beta_j = 0$ reveals that seed-guided words negatively influence the $\beta$ distribution, whereas $\beta_j = 1$ implies that the seed-guided words have an overinfluence in determining $\beta$ distribution. In this work, $\beta_j, \beta_k$ are set to 0.5, articulating that the influence of seed words and the general words have equal priority.

All the parameters and notations mentioned in Fig. 2 are listed in Table 1. In addition, the symbols and notations used throughout this work are also represented in Table 1 as a reference.

The sample algorithm for Fig. 2 to choose forensic relevant and irrelevant files is represented in Algorithm 1 with the inputs as: number of topics (K), relevant topics (R),

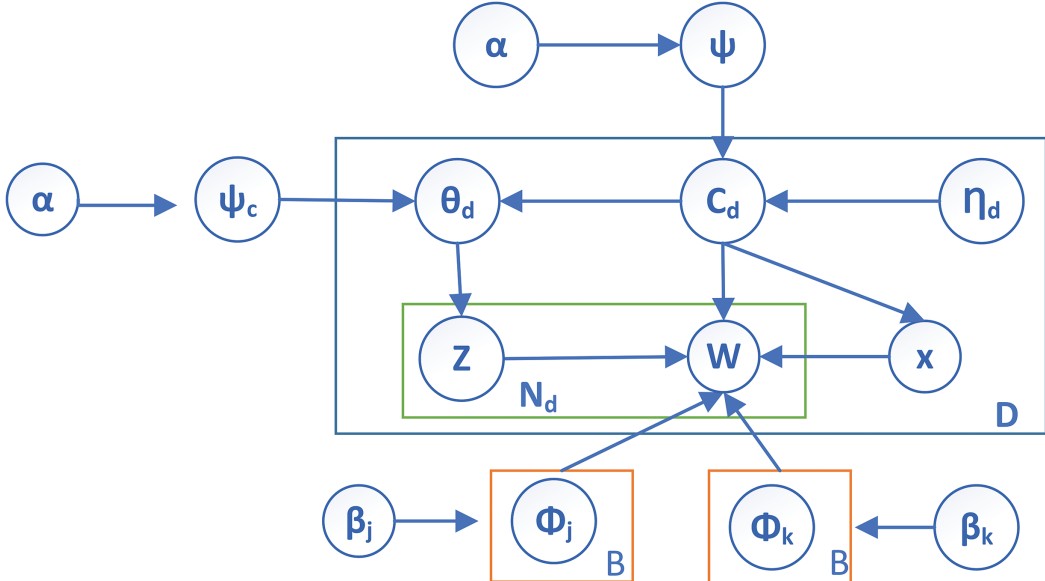

**Figure 2 Plate notation of proposed functional parameter in SFCS-$\beta_k$.**

seed words (S), hyperparameter for topic distribution ($\alpha$), word distribution within non-seed topics ($\beta_j$), seed topics ($\beta_k$), and scalar parameter ($\lambda_s$).

In guided LDA, many approaches have used the parameter $\beta$, which is used to adjust per-topic word distributions based on the seed words provided, and have set $\beta = 1$. In this work, the authors modified $\beta$ and introduced $\beta_k$ along with $\beta_j$ to use as seed words, such that $\beta = \beta_j \cup \beta_k$. The seed words in $\beta_j$ contain eight sets of forensic-related Blacklisted Keywords (BKW) defined by DHS, such as $\beta_k \in \{DS, H\&N, HC, IS, BV, T, D, CS\}$ where DS represents domestic security, $H\&N$ is nuclear information, HC is a disease and biological viruses, IS is infrastructure-related security, BV is violence-related words, T is terrorism-related words, D is emergency-related words, and CS is cybersecurity-related.

To incorporate seed information, a set of seed words $\beta_k$ over a set of topics $K$ is defined to modify the prior distribution over $\theta_d$. If $T_{ks}$ be the set of seed words, then $\theta_d$ is given as defined in the Eq. (17).

$$\theta_d \sim Dirichlet(\alpha + \lambda\beta_k * \mathscr{Y}_{k*\beta_k}) \tag{17}$$

where $\lambda\beta_k$ controls the seed information. $\mathscr{Y}_{k*\beta_k}$ is a binary vector that takes the value 0 for a topic $k$ if a $w$ is not found in $T_{ks}$ or takes the value as 1, if $w$ is found in $T_{ks}$. As a result, the prior probabilities for seed topics will be greater, as $\mathscr{Y}_{k*\beta_k}$ contains non-negative values.

In this work, the authors set $\beta = 0.01$ for words not in $T_k * \beta_k$ and $\beta = 1$ for the words that appear in $T_k * \beta_k$. This specifies that the topical model should give high probabilities to words that occur in $T_k * \beta_k$ and low probability of other words. These values are set according to the *corpus* being modelled. Equation (18) defines how seed words can be incorporated into the $\beta_k$ parameter.

**Table 1 List of notations.**

| Symbol | Meaning |
|---|---|
| $\alpha, \beta_j, \beta_k$ | Dirichlet priors |
| D | Total number of documents in the *corpus* |
| $\eta_d$ | Relevant-topic distribution of d |
| $\theta_d$ | General topic distribution |
| $\varphi$ | Ldir (Irrelevant topic distribution) |
| $\varphi_c$ | Prior topic distribution of $S_c$ |
| $C_d$ | Assignment of category-topic for each D |
| W | Size of vocabulary ($V_{oc}$) |
| Z | Topic assignment for word $w_{d,i}$ |
| s | Seed word |
| $S_c$ | Set of seed words |
| $\phi_k$ | Word distribution of general topics |
| $\phi_j$ | Category topic word-distribution |
| $w_{d,i}$ | Specific word at $i^{th}$ position |
| $z_{d,i}$ | Topic assignment for $w_{d,i}$ |
| B | Total number of general topics |

$$\beta = \begin{cases} 1, \text{if } w_i \in \beta_k \\ \beta_0, \text{otherwise} \end{cases} \tag{18}$$

where $\beta_0$ is the base value of $\beta$, $\beta_k$ is the set of seed words associated with the $k^{th}$ topic, and $w_i$ is the $i^{th}$ word in vocabulary.

From Eq. (18), when $w_i$ appears in $T_{ks}$ for a particular topic $k$, $\beta$ is set to 1, indicating that it strongly influences the generated topic.

To extract the seed words for $\beta_k$, a *Corpus* (C) containing a set of documents (D) with vocabulary size (V) and with words $(w_i, w_j, \ldots, w_n)$ must be converted into vectors for further computation. For this purpose, the authors used W2V with skip-gram model (*Rong, 2016*) and vectorised the data to capture the semantics of the context word within the focus word of the given radius. If $s$ represents the window size, $T$ denotes the sequence length, and $p$ indicates the position, then the likelihood function of the skip-gram is calculated as shown in Eq. (19).

$$\sum_{r=1}^{T} \sum_{-s \le p \le s, p \ne 0} \log P\left(w^{(r+p)} \mid w^{(r)}\right) \tag{19}$$

The context and word embedding layers are updated using the gradient descent method. Our work used the gradient descent method owing to its efficient optimization, highly flexible and customizable objective function, and high scalability, because our data is massive.

---

**Algorithm 1** Topic modeling algorithm.

---

**Input:** $\{K, S, R, \alpha, \beta, \beta_j, \beta_k, \lambda_s\}$

**Output**: Topic-word distributions: $\beta_k^{sk}$

  **for** $k \leftarrow 1$ to $K$ **do**

      Choose relevant topic $r$ from set $\{1, \ldots, R\}$ such that $\phi_{rk} \sim \text{Dir}(\beta_r)$

      Choose seed topic $\beta_k^{sk} \sim \text{Dir}(\beta_k^s)$

  **end**

  **for** $d \leftarrow 1$ *to* $D$ **do**

      Choose topic distribution $\theta_d \sim \text{Dir}(\alpha_0)$

      Draw irrelevant topic distribution $\phi_t \sim \text{Dir}(\alpha_1)$

  **end**

  **for** $w \in \{1, \ldots, D\}$ **do**

      Choose binary vector $\mathcal{Y}_{ks}$

      **if** $\mathcal{Y}_{ks} = 1$ *&& word* $w \in$ *set S* **then**

         Select topic $z_i \sim \text{Mult}(\theta_d)$

    **end**

    **if** $x_{d,w} = 0$ *&& word* $w \notin$ *set S* **then**

      Select topic $z_i \sim \text{Mult}(\theta_d)$

      Select word $w_{d,w} \sim \text{Mult}(\phi_{rz_i})$

    **end**

    **if** *word* $w \in$ *set S* **then**

      Select topic $z_i \sim \text{Mult}(\theta_d)$

      Select word $w_{d,w} \sim \text{Mult}(\beta_k^{S_{z_i}})$

    **end**

    Define seed per topic word distribution. $\beta_k^{sk} = (1 - \lambda_s) \cdot \beta_{ks} + \lambda_s \cdot \mathcal{Y}_{ks}$

  **end**

---

$$\textit{Softmax } J = -\log P(w_{c-s}, \ldots, w_{c-1}, w_{c+1}, \ldots, w_{c+s} \mid w_c)$$

$$= -\log \prod_{p=0, p \neq s}^{2s} P\left(w_{c-s+p} \mid w_c, u_{c-s+p} \mid v_b\right)$$

$$= -\log \prod_{p=0, p \neq s}^{2s} \frac{\exp\left(u_{c-s+p}^T v_b\right)}{\sum_{k=1}^{|V|} \exp\left(u_k^T v_b\right)} \tag{20}$$

$$= -\sum_{p=0, p \neq s}^{2s} u_{c-s+p}^T v_b + 2s \log \sum_{k=1}^{|V|} \exp\left(u_k^T v_b\right).$$

The time complexity for the above equation resulted in $O(|V|)$ as we have to compute all the inner products between the context word embedding and the centre word. Hence,

---

the Sigmoid function is used to optimise the time complexity and normalise the likelihood as $P(W_p = 1|w, c, \theta) = \sigma(v_b^T.v_w)$. The authors maximise $P(W_p = 1|w, c, \theta)$ as $\frac{1}{1+e^{(-v_b^T.v_w)}}$, where $(w, c)$ are word pairs in the training samples, $\theta$ is the model parameter, $W_p = 1$ represents word pairs from training examples, and $W_p = 0$ represents word pairs sampled randomly. Instead of sampling over positives to reduce time, word pairs can be negatively sampled from the *corpus* as shown in the Eq. (21).

$$
\begin{aligned}
\theta = \arg\max_\theta \ & \prod_{(w,c)\in W_p} P(W_p = 1 \mid (w, c), \theta) \\
& \times \prod_{(w,c)\in \tilde{D}} P(W_p = 0 \mid (w, c), \theta) \\
= \arg\max_\theta \ & \prod_{(w,c)\in W_p} P(W_p = 1 \mid (w, c), \theta) \\
& \times \prod_{(w,c)\in \tilde{D}} (1 - P(W_p = 1 \mid (w, c), \theta)) \\
= \arg\max_\theta \ & \sum_{(w,c)\in W_p} \log P(W_p = 1 \mid (w, c), \theta) \\
& + \sum_{(w,c)\in \tilde{D}} \log(1 - P(W_p = 1 \mid (w, c), \theta)) \\
= \arg\max_\theta \ & \sum_{(w,c)\in W_p} \log\left(\frac{1}{1 + \exp(-u_w^T v_b)}\right) \\
& + \sum_{(w,c)\in \tilde{D}} \log\left(\frac{1}{1 + \exp(u_w^T v_b)}\right).
\end{aligned}
\tag{21}
$$

Once the words are vectorised using the above equations, the authors identified all similar words by calculating the FCCS algorithm (*Xu et al., 2020*) as shown in Eq. (22). This algorithm, also known as the navigable small-world graph algorithm, is chosen for this work as it has considerable high-dimensional vectors.

$$
sim(p, x_i) = \frac{\sum_{n=1}^{N} <p, x_i>}{\sqrt{\sum_{n=1}^{N} p_n^2}\sqrt{\sum_{n=1}^{N} x_i^2}}
\tag{22}
$$

where $<p, x_i>$ is the dot product between $p$ and $x_i$, and $||p||$ and $||x_i||$ are the norms of $p$ and $x_i$, respectively. Based on this, the authors build an undirected graph with $n$ nodes, where n denotes $V_{x_i}$ such that the cosine similarity is measured among the edges between the nodes, represented as G = (V, E), where $V = x_1, x_2, ..., x_n$ and E = $(x_i, x_j)|i \neq j, cos\_sim(x_i, x_j) \geq \varepsilon$, where $\varepsilon$ is the adjusted similarity threshold. We compute the distance to each query vector $q$ for a node $v$ as $d(q, v) = cos\_sim(q, v)$. The similar neighbours with high cos_sim scores are chosen from the current node $c$ to $q$ by the Eq. (23).

$$
c' = \arg\max\_\{v \in N(c)\} \cos \_sim(q, v)
\tag{23}
$$

where $N(c)$ is the set of neighbours of $c$ in (V, E).

The high-similarity words identified by domain experts in the previous work are denoted as common seed word sets and represented as $\beta_j$. The $\beta$ parameter is now constructed as $\beta = \beta_j \cup \beta_k$ that contains efficient seed words. Based on these seed words, the LDA model is guided, and the resultant topics are stored in a central repository for further processing. The proposed algorithm for extracting relevant seed words with documents (D), relevant topics (R), irrelevant topic (I) is given in Algorithm 2.

## Keyword-metadata-pattern classifier

The Keyword-Metadata-Pattern (KMP) classifier is used in the author's existing work to identify forensic-relevant files. However, incorporating the $\beta_k$ parameter can still improve the current system regarding forensic file identification and classification. Therefore, the proposed work includes the KMP classifier at the tier-2 level for preliminary classification. Based on the results of $\beta_k$ in tier-1, keywords are collected, indexed, and stored in a central repository which serves as an input to the Blacklisted keyword search module of tier-2. If any document $D$ matches any of the indexed words, that document is flagged as a relevant file, and the remaining documents are irrelevant in the other case. The condition to flag a file as sensitive or not is defined in the Eq. (24).

$$\prod_{v_i=0}^{\max(V_i)} \left\{ \begin{array}{l} v(v_i, d_i) = 1, \ \text{if } v \in D \\ v(v_i, d_i) = 0, \ \text{if } v \notin D \end{array} \right\}. \tag{24}$$

The relevant files are extracted based on the above equation and the remaining two modules in tier 2, such as Metadata and Pattern. Finally, the files are preliminarily classified as relevant and irrelevant based on the tier-2 results, and thus, unstructured data is transformed into structured data. To classify the files automatically, the proposed model is trained with L-SVM given as $K(x, x') = <x, x'>$, where $x$ and $x'$ are two input vectors that can be represented in a hyperplane, and $<>$ represents the dot product operation of the two vectors. The hyperplane for the L-SVM is defined as $w^t x + b = 0$ where $w$ is the weight vector perpendicular to the hyperplane, $b$ is the bias term. Therefore, the decision function for L-SVM can be represented as $f(x) = sign(w^t x + b)$ where sign returns $-1$ if the argument is irrelevant and returns $+1$ if the argument is relevant. In other words, the files are classified as relevant or irrelevant based on the Eq. (25).

$$f(x_i) = \left\{ \begin{array}{ll} -1 & if \ <w, x> \ + \ b \geq 0 \\ +1 & if \ <w, x> \ + \ b < \ 0 \end{array} \right. . \tag{25}$$

Compared with other kernels such as the radial basis function (RBF) and polynomial, the linear kernel has more advantages in terms of training time, ability to fit, low risk of over-fitting, and no cost for hyper-parameter tuning. Furthermore, a Linear SVM is less prone to overfitting, results in high accuracy when data is linearly separable, and is highly efficient when working with two classes.

**Algorithm 2 SFCS-$\beta_k$ seed topic extraction algorithm.**

**Input:** {D, R, I}

**Output:** $T : <w\_1, w\_2, \ldots w\_n>$

**for** $d \in D$ **do**

    create empty list $L_d^r$ for relevant topic r

    **for** $i \in d$ **do**

        calculate topic likelihood score $P(Z_{d,i} = r; X_{d,i} - 0 | C_d = r)$

        Sample $Z_{d,i}$ and $X_{d,i}$ based on $L(\frac{\theta}{x})$

        $L_d^r.append(Z_{d,i}, X_{d,i})$

    **end**

**end**

**for** $i_r \in$ **do**

    create empty list $L_d^{i_r}$ for irrelevant topic I

    **for** $i \in d$ **do**

        calculate topic likelihood score $P(Z_{d,i} = i_r; X_{d,i} - 0 | C_d = i_r)$

        Sample $Z_{d,i}$ and $X_{d,i}$ based on $L(\frac{\theta}{x})$

    **end**

**end**

**if** $c_d == r$ **then**

    $L_d = L_d^r$

**end**

**if** $c_d == i_r$ **then**

    $L_d = L_d^t$

**end**

$L_d^r.append(Z_{d,i}, X_{d,i})$

$\beta = \beta_j \cup L_d^r.$

# EXPERIMENTAL RESULTS AND ANALYSIS

The experiments in this work are evaluated on a Windows 11 X64 environment with 16 GB DDR4 RAM, i7 12700H 12-core CPU, and 8 GB DDR6 Nvidia RTX 4050 with 2,560 cores of 1.76 GHz. Keras and TensorFlow with Python are used in this work as backend servers. The datasets used in this work are GovDocs (*Garfinkel et al., 2009*), Real Drive corpus (RDC) (*Garfinkel et al., 2009*), and MSX-13 *Corpus* (*Roussev & Quates, 2013*) which consists of nearly 1 million raw documents. These documents include 8,014 raw word document processing files, 6,461 text files, 1,124 pdf files, 146 rich text format files, and 2,474 html files in addition to the other files. Document-related files in the dataset are pre-processed with the help of the Natural Language Toolkit (NLTK) library by

implementing tokenization, stop-word removal, and lemmatization. All the datasets and codes used in this work are available at DOI:10.5281/zenodo.8163855.

These words in the *corpus* are converted into vectors using the W2V (*Rong, 2016*) represented in Eq. (19). RDC consists of 18.3 million tokens, of which 0.15 million are unique. When the data is pre-processed, 10.9 million tokens are removed and converted to vectors using the W2V model with a dimensionality of 300 and a batch size of 1,000, with each epoch considering approximately 5,000 batches. The time taken for vectorization is approximately 6.5 hours on an 8 GB DDR6 Nvidia RTX 400 and 16 GB DDR4 RAM using TensorFlow 2.16.1 in a Python environment. PEM extracted 3.8 K words under the $\beta_j$ parameter, while 4.5 K relevant words are identified by the $\beta_k$ parameter. The $\beta$ parameter is now constructed using $\beta = \beta_j \cup \beta_k$ that contains 0.7 K seed words under eight categories defined by DHS. The TM algorithm is now guided over the $\beta_k$ parameter, and the resultant topical words are fetched. The proposed work is compared with Multidisk-LDA (M-LDA) (*Noel & Peterson, 2014*), P-LDA (*Gupta & Katarya, 2021*), SGLDA (*Li et al., 2018*), and SDOT (*Joseph & Viswanathan, 2023*) in terms of seed word extraction, and the results are presented in the following tables.

The proposed model SFCS-$\beta_k$ is compared with the existing models, such as M-LDA, which uses $\alpha$ (*Noel & Peterson, 2014*); SGLDA that uses $\alpha, \beta, and \beta_j$ (*Li et al., 2018*); P-LDA that uses $\alpha, \beta$, and $\beta_j$ (*Gupta & Katarya, 2021*); and SDOT that uses $\alpha, \beta$, and $\beta_j$ (*Joseph & Viswanathan, 2023*) hyperparameters. Since P-LDA (*Gupta & Katarya, 2021*) was trained explicitly on the pandemic dataset, the results of this model are less thematic than those of the other models. Even though few works seem outdated or trained explicitly on different datasets, the authors used them in this work as they are closely aligned. It should be noted that M-LDA (*Noel & Peterson, 2014*) is the first work to introduce LDA to the DF domain, followed by SDOT (*Joseph & Viswanathan, 2023*). Apart from these two works, P-LDA (*Gupta & Katarya, 2021*) also introduced $\beta_j$ parameter to the LDA algorithm; therefore, this work is also evaluated along with the proposed model.

Table 2 represents the extracted words for the topic 'terrorism'. Under this category, existing methods could identify only similar words, whereas the proposed approach extracted terrorism-related words, including terrorist organisations.

Tables 3 and 4 summarise the results of the extracted keywords for the topics 'alqaeda' and 'forensic investigation'. The proposed method identified significant terrorist organisations along with characteristics associated with Al Qaeda, whereas existing systems could only identify characteristics. Under the topic 'forensic investigation', existing methods extracted actions of the topic, whereas the proposed method identified the process along with the topic's actions. The results suggest that the proposed method extracted words based on contextual and semantic meaning, while the existing methods could only extract the actions or results associated with the topics.

By integrating the PEM module into the BKW module, a remarkable 5.6 K sensitive keywords have been successfully identified from 59.9 K files. The tier-2 outcomes were subsequently utilized to accurately label and categorise the files into two categories: relevant and irrelevant. Now, the system is trained with linear SVM for automated file classification with *feature_size* $(30, 35, 40, 45, 50)$, and the proposed system classified the

**Table 2 Comparative analysis of the keywords extracted for the topic 'terrorism'.**

| SGLDA $\alpha, \beta, \beta_j$ (Li et al., 2018) | M-LDA ($\alpha$) (Noel & Peterson, 2014) | P-LDA ($\alpha, \beta, \beta_j$) (Gupta & Katarya, 2021) | S-LDA ($\alpha, \beta, \beta_j$) (Joseph & Viswanathan, 2023) | Proposed work SFCS-$\beta_k(\alpha, \beta, \beta_j, \beta_k)$ |
|---|---|---|---|---|
| Attack | Violence | Militancy | Terrorist | Jihad |
| Fear | Terrorism | Attack | Radicalism | Sleeper cells |
| Security | Suicide | Violence | Violence | Insurgency |
| Hostage | Bombing | War | Bomb | Suicide |
| Propaganda | Ideology | Terrorism | Sabotage | Counterterrorism |
| Ideology | Recruitment | Suicide | Cyberwar | Intelligence |
| Political violence | Propaganda | Subversion | Attack | Isis |
| Terrorism | War | Dissent | Hezbollah | Militancy |
| Extremism | Insurgency | Revolt | Suicide bomb | Radicals |
| Violence | Fundamentalism | Bio war | Militancy | Bombing |

**Table 3 Comparitive analysis of the keywords extracted for the topic 'alqaeda'.**

| SGLDA ($\alpha, \beta, \beta_j$) (Li et al., 2018) | M-LDA ($\alpha$) (Noel & Peterson, 2014) | P-LDA ($\alpha, \beta, \beta_j$) (Gupta & Katarya, 2021) | S-LDA ($\alpha, \beta, \beta_j$) (Joseph & Viswanathan, 2023) | Proposed work SFCS-$\beta_k(\alpha, \beta, \beta_j, \beta_k)$ |
|---|---|---|---|---|
| Bloodshed | Islamic state | Terrorist attack | Terrorism | Isis |
| Global threat | Terrorist group | Car bomb | Suicide | Hezbollah |
| Afghanistan | Islamist | Jihad | Bio war | Boko haram |
| Hatred | Harm | Terrorism | Threat | Islamic state of iraq |
| Ideology | Extremist | Suicide | Sharia | Al-Zawahiri |
| Salafi | Terrorism | War | Insurgency | Bin laden |
| Jihad | Radical | Muslim brotherhood | Isis | Lashkar-e-taiba |
| Mujahideen | Jihad | Threat | Fundamentalism | Hamas |
| Smuggling | Illegal | Prohibited | Pakistan | Suicide bombing |
| Sharia law | Violence | Extremism | Afghanistan | 9/11 attack |

**Table 4 Comparitive analysis of the keywords extracted for the topic 'forensic investigation'.**

| SGLDA ($\alpha, \beta, \beta_j$) (Li et al., 2018) | M-LDA ($\alpha$) (Noel & Peterson, 2014) | P-LDA ($\alpha, \beta, \beta_j$) (Gupta & Katarya, 2021) | S-LDA ($\alpha, \beta, \beta_j$) (Joseph & Viswanathan, 2023) | Proposed work SFCS-$\beta_k(\alpha, \beta, \beta_j, \beta_k)$ |
|---|---|---|---|---|
| Arrest | Justice | Investigation | Case analysis | Autopsy |
| Crime | Law | Court | Expert | Investigation |
| Fraud | Toxicology | Acquisition | Forensic science laboratory | Evidence |
| Burglary | Corruption | Scene reconstruct | Fingerprint | Witness |
| Smuggling | Crime | Suspect | Legal | Prosecution |
| Ballistics | Analysis | Case analysis | Evidence | Expert |
| Scene | Court | Pathology | Evidence analysis | Court |
| Court | Trace | DNA analysis | Court | Suspect |
| Expert | Suspect | Toxicology | Fraud | Evidence |
| Evidence | Witness | Forensic analysis | Pathology | Forensic |

files with 94.6% accuracy. We used precision, recall, f1-score, accuracy, and Matthew's Correlation coefficient (MCC) to check the proposed model's performance.

Figure 3 shows the performance metrics of the proposed model for five feature sizes ($f_{size} = \{0.30, 0.35, 0.40, 0.45, 0.5\}$). The dataset splitting is considered randomly afresh each time to ensure diverse representation, reduce overfitting, and ensure the model's performance consistency. Even though the results of considered features do not significantly impact, the authors presented results for x = 35 alone for the reader's understanding and flexibility. To project the results more clearly, this work considered y-axis values in the range of $N \in \{89 : 98\}$ for Fig. 3.

SFCS-$\beta_k$ is compared with the author's previous work and existing models in performance metrics such as precision, Recall, F1-measure and Accuracy. The comparison result is represented using a box and whisker plot and is shown in Fig. 4. This plot measures the skewness in the distribution and any outliers, if present. Even though the dataset is discrete, the authors used this plot to visualize data values spread.

A few performance metrics used in this work are explained below.

## Performance metric evaluation

The proposed model is evaluated using precision, recall, ROC curve, F-measure, accuracy, and confusion matrix. Furthermore, to measure the model's robustness and performance, SFCS-$\beta_k$ is evaluated using the precision-recall (PR) curve and Matthew's correlation coefficient (MCC), as mentioned below.

*Reciever-operating characteristic curve (ROC)*: A ROC curve is created by plotting the true positive rate (TPR) and the false positive rate (FPR) at the possible thresholds. It targets TPR against FPR by showing the performance of a binary classifier, where each point on the curve depicts the threshold value. ROC curve is illustrated in Fig. 5, where FPR is plotted on the y-axis, and TPR is plotted on the x-axis.

*PR curve*: A PR curve is created by plotting the precision and recall of a single classifier at different thresholds, illustrating a negative slope function. A PR curve targets the minority class, whereas an ROC curve covers both minority and majority classes. Since our data is imbalanced, considering accuracy alone misleads the classification results. Therefore, to assess the precise performance of the actual class, we used the PR curve in this work. The area under PR-curve can be calculated using Eq. (26) and represented the same in Fig. 6.

$$AUC_{PR} = \sum_{i=1}^{n-1} \frac{(P_{i+1} - P_i)(R_{i+1} + R_i)}{2} \tag{26}$$

where $n$ is total threshold number, $P_i$ is the precision at $i^{th}$ threshold, and $R_i$ is the recall at $i^{th}$ threshold.

*Confusion matrix*: A Confusion matrix is a performance metric evaluator that visualizes the quality of the classification model by predicting the true positives and true negatives. Furthermore, the Confusion matrix shows the model errors by calculating false positives and false negatives. One can calculate the quality metrics like precision and recall using the above metrics. The proposed work's performance is evaluated based on a confusion

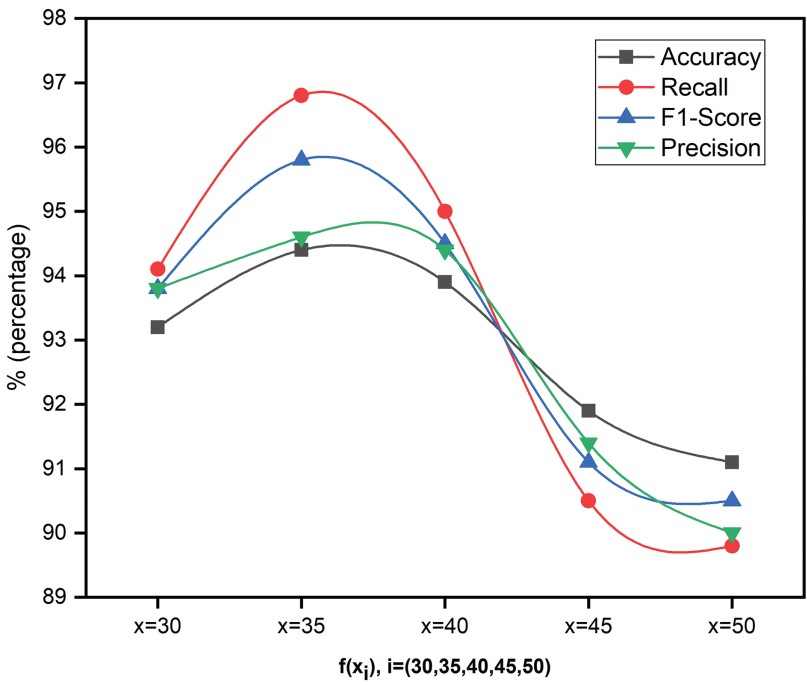

**Figure 3** Performance analysis of SFCS-$\beta_k$ using different levels of features.

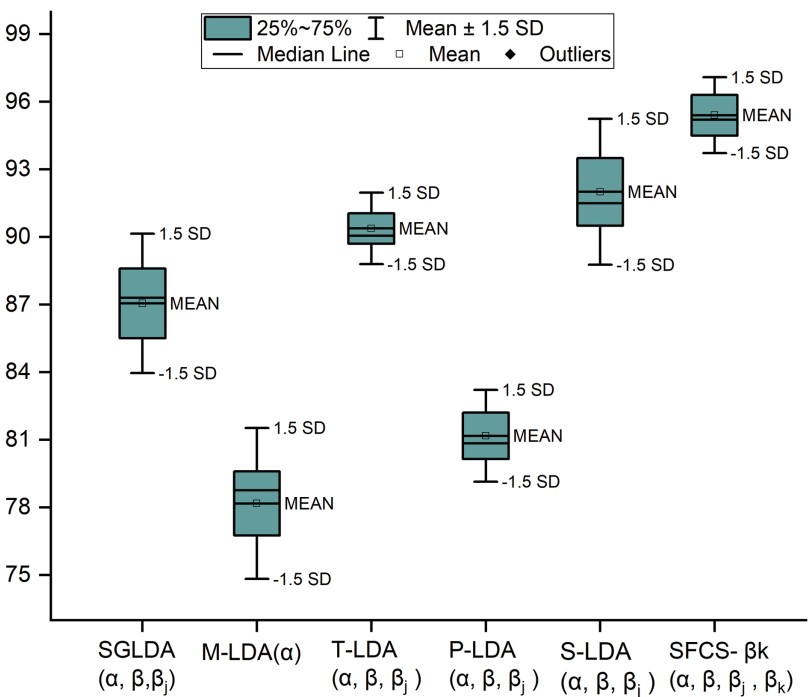

**Figure 4** Comparative performance analysis of SFCS-$\beta_k$ with existing models.

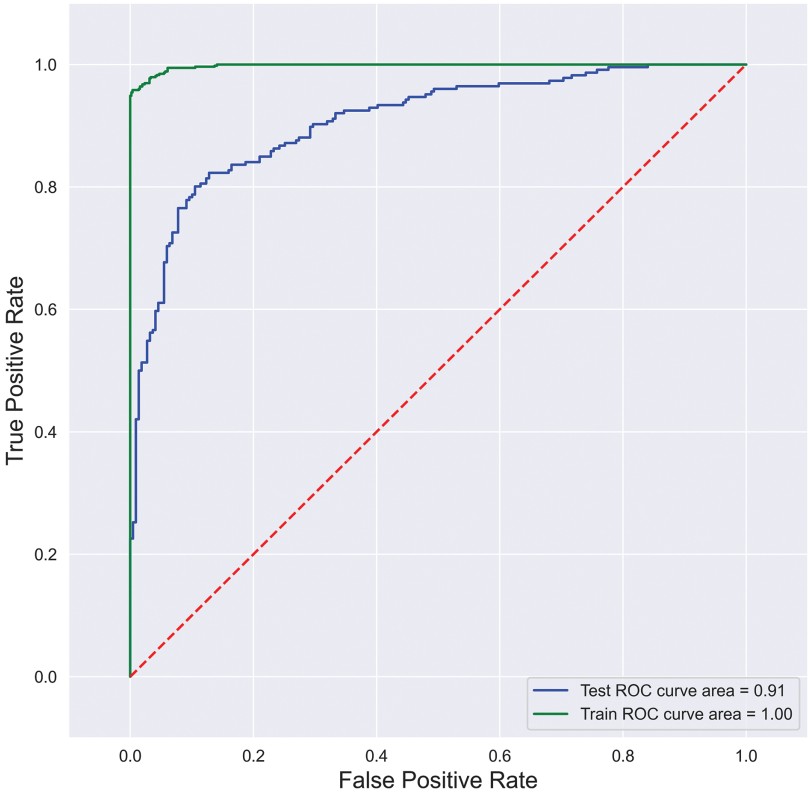

**Figure 5** SFCS-$\beta_k$ classification performance with AUC-ROC curve.

matrix, where predicted labels are shown on the x-axis, and actual labels are shown on the y-axis. and is shown in the Fig. 7.

*Matthew's correlation coefficient (MCC)*: MCC is a statistical metric used to evaluate the relation between the actual class-1, and predicted class-1 values within the range of (−1 to 1). MCC can be calculated by using the Eq. (27)

$$MCC = \frac{(T_P \times T_N) - (F_P \times F_N)}{\sqrt{(T_P + F_N)(T_P + F_P)(T_N + F_P)(T_N + F_N)}} \tag{27}$$

where $T_p$ is true positive, $T_n$ is true negative, $F_p$ is false positive, $F_n$ is false negative.

### Time complexity evaluation

The proposed model has also been assessed for its time complexity, with a breakdown provided for each phase.

1. Loading a pre-trained model assumes that a *corpus* with D documents requires O(n log n) time, where n represents the number of words.

2. Computing the cosine similarity for each target word with a context word requires O(V), which is computationally expensive.

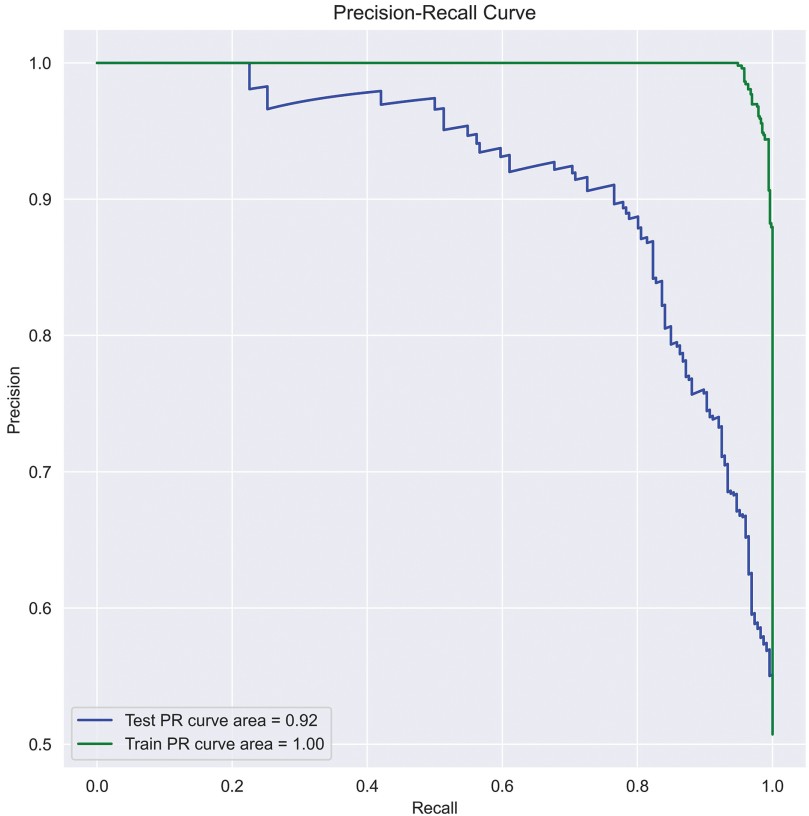

**Figure 6** SFCS-$\beta_k$ classification performance with PR-curve.

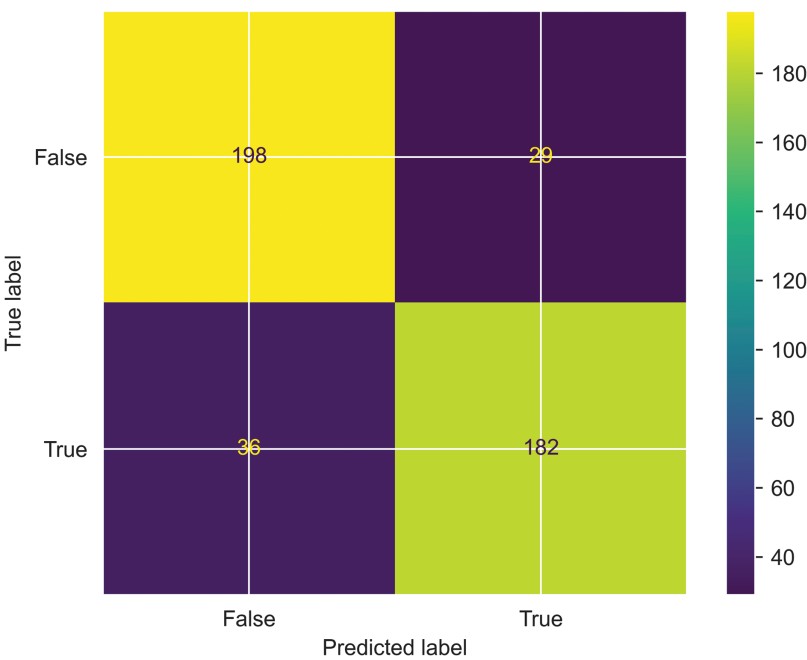

**Figure 7** Evaluation of SFCS-$\beta_k$ classification performance based on Confusion matrix.

3. To overcome this, the authors used the FCSS algorithm that takes $O(V * logV + V * q * D * \log q)$ time, where V represents the vocabulary size, D is the dimensionality of the vectors, and q is the nearest neighbour.

4. Training the guided LDA model with the proposed parameter requires a time complexity of $O(T * K * N * D)$, where T is the number of iterations, K is the number of topics, and N is the vocabulary size.

5. To train the SVM model with linear kernel, it takes $O(n * d)$, where $n$ is the number of samples, and $d$ is the number of features. Since $d$ is relatively small in the current context, linear SVM outperforms SVM with optimisation algorithms.

Finally, the overall system took $O(n \log n)$ time complexity, which is an optimal result compared to existing systems.

## DISCUSSION

This research establishes a strong correlation between seed-guided words and forensic investigations that identify interesting and uninteresting files. This research also investigated the seed words' influence on the classification model and how the relevant extracted seed words speed up the investigations. The proposed model is compared with the baseline models in terms of statistical, mathematical, and experimental evaluation that uses different hyperparameters for file classification. Even though existing models extract seed words, they still fall short of identifying polysemy and contextual seed words, which is achieved in this study. The comparative results presented in Tables 2–4 justify the contextual analysis as well as the influence of the forensic relevant seed words. To train the proposed model with marginal FP rates, this research considered four feature sets where the test data is divided into 30%, 35%, 40%, 45% and 50%. Furthermore, this research has hyper-tuned the parameters discussed in "Role and Tuning of Hyper Parameters" and set the values as $\alpha = 0.7$, $\beta_j = 0.09$, and $\beta_k = 0.5$. Upon optimizing the parameters, the classification model generated the best results when the training data was 65%, test data was 35%, and validation data at 15%, which is represented in Fig. 3. Furthermore, the proposed model's performance is compared with baseline models in descriptive statistics to show the distribution of skewness and the numerical data regarding minimum score, lower quartile, upper quartile, median and maximum score. This comparison is illustrated in Fig. 4; the following are observations.

1. When the standard deviation (SD) is set to 1, outliers emerge. When SD = $\pm 1.5$, outliers are not detected. Moreover, 93% of data is distributed within this range.

2. The median of a box plot lies outside of the box of a comparison box plot, which shows the difference between the two groups.

3. When interquartile ranges are compared to examine the data dispersion between each sample, data dispersion is less for the proposed work, whereas dispersion is relatively high for other boxes.

4. By observing the extreme ends of the two whiskers, it can be seen that the proposed system has a shorter range, indicating that the data is less scattered. In contrast, the other systems have a higher range, resulting in scattered data.

5. Finally, we can understand that the proposed system has a normal distribution, whereas other distributions tend towards positive and negative skew.

Furthermore, to analyze the model's robustness and the degree of classification, this research considered the AUC-ROC curve shown in Fig. 5. The area under the curve is ≥0.85%, representing an 85% probability of correctly classifying a random positive and negative example. Further performance metrics reveal that the proposed model can handle extensive data with the best classification rates.

### Error analysis

The authors considered error analysis to systematically analyze the errors made by the proposed classification model in order to understand its limitations and for further improvements. For this, the authors considered the instances where the proposed model predicts a relevant class incorrectly and the instances where an irrelevant category is mispredicted. As a result, a confusion matrix is constructed to visualize the incorrectly identified classes. This is due to the impact of support vectors on decision boundaries. Therefore, the authors implemented the hyperplane maximization technique by optimizing the objective of SVM as shown in Eq. (28).

$$\min \frac{1}{2}||w_v||^2 + L \sum_{i=1}^{n} \xi_i \tag{28}$$

where $w_v$ is the weight vector perpendicular to the hyperplane (H),

$||w_v||^2$ is the Squared normalization of the weight vector ($w_v$),

$L$ is the regularization parameter that controls the bias between the margin maximization and classification error,

$n$ is the total number of samples used in training, and

$\xi_i$ is the slack variable for representing inaccurate classifications.

To ensure the correct classification of the data points on either side of the hyperplane, the authors implemented SVM constraints as mentioned in Eq. (29).

$$y_i(w^T x_i + b) \geq 1 - \xi_i \quad \text{for all } i = 1, \ldots, n \tag{29}$$

where $b$ is a bias term,

$y_i$ is a label that represents either +1 or −1 as $i$, and

$x_i$ is a feature vector for $i$.

Thus, by optimizing the hyperparameters in LDA and SVM, the proposed model can overcome misclassification errors, proving its reliability and robustness. Therefore, an investigator can now determine the file relevancy based on the relevant seed words and present evidence as admissible in court. Moreover, the proposed model can be integrated into digital forensic frameworks to identify interesting files and expedite investigation. However, this work can be trained using long short-term memory (LSTM) or neural

networks in the near future to reduce the time frame, which can significantly enhance the current work.

## CONCLUSION

The SFCS-$\beta_k$ system, designed specifically for the forensic domain, incorporates domain-specific seed words and employs optimization techniques for forensic file classification. The SFCS-$\beta_k$ system effectively reduces forensic investigation latency by examining forensic relevant files within a large *corpus*, utilizing the introduced parameter $\beta_k$. Furthermore, this study effectively addressed challenges like topic sparsity, model overfitting and suboptimal topic distribution when $\alpha, \beta$, and $\beta_j$ parameters are used. The proposed $\beta_k$ parameter is incorporated into an automated parametric extraction module in tier-1 to extract seed words using semantic and contextual similarity in vector space. This extraction is achieved by integrating Word-to-Vector with the LDA algorithm by considering textual, lexical, and forensic domain-specific features. This integration builds upon the author's prior work, which focused on optimizing forensic file labelling and classification techniques. Furthermore, to enhance the classification rate, this study implemented the hyperplane maximization technique and the optimization of hyperparameters. The results indicate that the proposed parameter $\beta_k$ has successfully detected 700 blacklisted keywords and flagged 5.6 k files as suspicious out of 59.9 k files in RDC. Furthermore, the proposed system successfully removed 278,000 forensic irrelevant files from the *corpus*, resulting in a notable decrease in computation time. Consequently, the time complexity is reduced from $O(\text{n log n} * |V|)$ to $O(\text{n log n})$, resulting in faster prediction and effective handling of extensive data. The proposed work was compared with state-of-the-art models, such as SG-LDA, Tag-LDA, Multidisk-LDA, P-LDA, and SDOT which used $\alpha, \beta$, and $\beta_j$ parameters for topic distribution. In comparison to the existing models, the SFCS-$\beta_k$ system demonstrated a file classification accuracy of 94.6%, precision of 94.4%, recall of 96.8%, and F1-score of 95.8% surpassing all the previous LDA models. The proposed methodology represents a pioneering effort to incorporate the guided LDA model within the digital forensic field. Thus, it serves as an invaluable guide for the development of effective DF-guided models. The author's future work involves training and implementing a large-scale forensic file classification based on neural networks or LSTM, which could enhance the accuracy and classification rate besides reducing the investigation time.

### Funding

Vellore Institute of Technology, Vellore, India supported the APC for this article. The funders had no role in study design, data collection and analysis, decision to publish, or preparation of the manuscript.

## Grant Disclosures

The following grant information was disclosed by the authors:
Vellore Institute of Technology, Vellore, India.

## Competing Interests

The authors declare that they have no competing interests.

## Author Contributions

- D. Paul Joseph conceived and designed the experiments, performed the experiments, performed the computation work, prepared figures and/or tables, and approved the final draft.
- Viswanathan Perumal conceived and designed the experiments, analyzed the data, authored or reviewed drafts of the article, and approved the final draft.

## Data Availability

The data is available at Zenodo: Paul Joseph. (2023). Hypertuned Beta parameter SDOT Framework for Robust Forensic File Classification [Data set]. Zenodo. https://doi.org/10.5281/zenodo.8163855.

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
