# Peer review of "Optimizing forensic file classification: enhancing SFCS with βk hyperparameter tuning"

_PeerJ Computer Science, doi:10.7717/peerj-cs.2608_

## Round 0.1 · original submission · Major Revisions

Dear authors,

Thank you for submitting your article. Feedback from the reviewers is now available. It is not recommended that your article be published in its current format. However, we strongly recommend that you address the issues raised by the reviewers, especially those related to readability, experimental design and validity, and resubmit your paper after making the necessary changes.

Best wishes,

·

Basic reporting

The manuscript generally uses professional English, but there are instances of grammatical errors and awkward phrasing that could be improved. Consider revising complex sentences for clarity and engaging a fluent English speaker or a professional editor to proofread the document. The literature review provides a good overview, but it could benefit from a more detailed comparison with existing methods to better highlight the study's novelty and significance. The article structure conforms to academic standards, but the quality of some figures and tables needs improvement. Ensure all figures are high resolution and tables are clearly labeled with explanatory notes. Raw data is shared, which is commendable, but ensure it is easily accessible and well-documented for replication purposes. The results are relevant to the hypotheses, but a more in-depth analysis and discussion linking findings to research questions is needed. Additionally, include clear definitions of all terms and theorems used in the study and ensure that any provided proofs are detailed and easy to follow for readers.

Experimental design

The manuscript presents original primary research within the journal's aims and scope, and the research question is well defined, relevant, and meaningful. It successfully identifies and addresses a significant knowledge gap in the field of forensic file classification. However, the manuscript could benefit from providing more detailed descriptions of the experimental setup, including hardware specifications, software used, and the duration of experiments, to ensure reproducibility. The process of hyperparameter tuning should be elaborated upon, explaining how specific values were chosen and their impact on model performance. While the methodology section is comprehensive, including more detailed descriptions of the algorithms used and the steps involved in each stage of the proposed framework would enhance clarity and replicability. Control experiments should be included to demonstrate the baseline performance against which the proposed method is compared. Overall, the investigation is rigorous and meets high technical and ethical standards, but adding these details would further strengthen the manuscript.

Validity of the findings

The manuscript does not explicitly assess the impact and novelty of the findings. It is important to clearly articulate the significance and potential contributions of the proposed method to the field of forensic file classification. The rationale and benefits of the study should be explicitly stated to encourage meaningful replication. While all underlying data have been provided and appear robust, statistically sound, and controlled, the manuscript would benefit from a more detailed analysis of the statistical significance of the findings. Conducting statistical tests to determine the significance of the improvements observed in the proposed method compared to existing methods would strengthen the validity of the results. The conclusions are well stated and linked to the original research question, but they should be more explicitly tied to the supporting results with a detailed discussion on the implications of the findings. Additionally, including an error analysis section to discuss potential limitations or sources of error would provide a more comprehensive understanding of the study's validity.

Additional comments

1. Elaborate on the process of hyperparameter tuning, including specific ranges and values considered for each parameter, and the criteria used to select the optimal values.
2. Provide detailed information on the hardware and software environment used for the experiments, including the specifications of the machines and versions of the software libraries.
3. Include more detailed comparisons with baseline methods, highlighting the differences in performance and explaining why the proposed method outperforms them.
4. Describe the data preprocessing steps in greater detail, including any normalization, data cleaning, or augmentation techniques used.
5. Offer more detailed descriptions of the algorithms used in each stage of the proposed framework, including pseudocode or flowcharts to illustrate the processes.
6. Justify the choice of evaluation metrics and explain how they are calculated. Discuss why these metrics are appropriate for assessing the performance of forensic file classification systems.
7. Conduct statistical tests (e.g., t-tests, ANOVA) to determine the significance of the improvements in performance metrics and include these results in the manuscript.
8. Provide more visual examples of the model's performance, such as confusion matrices or ROC curves, to better illustrate its effectiveness.
9. Discuss potential limitations of the proposed method and how they might be addressed in future work.
10. Include a section on the computational complexity of the proposed method, comparing it with baseline methods and discussing its efficiency.
11. Explain the rationale behind the selection of datasets used in the experiments and provide more details about their characteristics and relevance to the study.
12. Discuss the scalability of the proposed method and its applicability to larger datasets or different forensic scenarios.
13. Provide more detailed explanations of the feature extraction process, including the specific features used and why they were chosen.
14. Include a more thorough error analysis, discussing common misclassifications and potential reasons for these errors.
15. Clarify the role and impact of each hyperparameter in the proposed method, providing intuition on how they affect the model's performance.
16. Discuss the potential real-world applications of the proposed framework and its implications for forensic investigations.
17. Include a discussion on the ethical considerations of using AI in forensic file classification, particularly regarding privacy and data security.
18. Ensure that all figures and tables are high resolution and clearly labeled, with detailed legends explaining their significance.
19. Provide links to code repositories and datasets used in the study to facilitate replication and further research.
20. Summarize key findings and their implications in the conclusion, highlighting the main contributions and suggesting specific areas for future research.

Cite this review as

Reviewer 2 ·

Basic reporting

Abstract should be compact not more than 15 lines.
Make References point wise

Experimental design

technical and ethical standards were maintained properly

Validity of the findings

Conclusion mentioned is well stated

Additional comments

nil

·

Basic reporting

Abstract: Need Revision
Keywords: Good
Introduction: Average/Revised Please
Literature Review: Average
Data Set: Average
Methodology: Weak
Caption, Citations & Footnotes: Good
Pictures, graphs & Flowcharts: Average
Results: Weak
Conclusion: Average
Future Work: Poor
References: Average
----------- Overall evaluation -----------

Experimental design

The authors present a technical research paper with relevant topic, proper research methodology and potentially good contribution to the field of studies.
The authors are encouraged to resubmit the paper with more clarity on presented performance assessment metrics with the selected relevant Case studies and possible application scenario with assessment metrics. The paper should be written in proper format, figures should fit within the text, use of font should be uniform in all paper, as well as references should be updated with most recent results.

Suggestion and Recommendation:
1. Authors may elaborate more on the novelty/contribution of their work and how it
2. Authors need to be specific about their problem statement and the scope of their research.
3. Abstract: elaborate more on the problem statement, findings, and contributions.
4. Introduction is not clear. Authors may contribute more towards this.
Contributes to the literature in the second last paragraph of the introduction clearly.
5. Thorough proofreading is recommended.
6. A few of the figures are taken from the sources and are not cited properly, either they may be cited properly with permissions or may be removed/ redrawn.
7. The conclusion is not clear and needs revision and clarity and alignment with the abstract and title.

Validity of the findings

1. In the introduction, the scientific problem of the existing evaluation is missing. There should initially be discussed the actual problem and then the research motivation.
2. Please highlight major contributions of this work in this current version, otherwise the current form shows weak/lack of novelty.
3. Please refine the language of this paper, such as avoid we, they, our, and other related words in this paper.
4. Please improve the portion of problem description and problem formulation of the proposed work. Cannot find novelty in the current form.

Additional comments

References:
1. Your references are not listed in good style, as citation style is different from one paper to other.
2. some of your references are not complete please check.
3. Some citations (references) created in wrong manner (Please follow journal's criteria).

Authors are encouraged to base on recent references about the current development in blockchain technology. Moreover, technology collaborates with other technologies to create new paradigms, such as artificial intelligence, such machine learning, deep learning, with federated learning.

Reviewer 4 ·

Basic reporting

The manuscript presents an additional parameter for finetuning and extracting relevant keywords in a text classification problem. The proposed parameter extends the common body of knowledge to include topical relevance in the search space.
A detailed and rigorous presentation of the proposed parameter is presented in the manuscript with verifiable results. Overall, the manuscript is well written. However, the following can be considered going forward:
1. the stated research questions on pages 3/18 lines 102-104 have not been answered in the result or the discussion section of the manuscript. I would recommend a well-defined research question(s) in line with the study.
2. It is not clear how this study can be defined as a forensic scope. Whilst the research question on page 3/18 line 104 appears to relate to forensics, the problem being addressed is not necessarily a forensic problem but a threat intelligence problem in general. I would therefore recommend that the author consider the wider implications of the study, limiting it to forensics. Yes, the dataset used can be from a forensic source, but that is not to limit the study. Other similar threat-hunting and incident response processes can rely on the proposed topical and contextual relevance proposed by the introduced parameter.
3. The study presented the area under the PR curve only, with a justification for the same. However, as also identified in the study on page 15/18 in Line 412, the ROC also provide efficient clarity. I would therefore suggest that the result of the AUC be included in the revision.
4. The lack of a discussion contributed to the limitation of the study. Perhaps that would have been a section to link the research questions to the results obtained.
5. A minor observation, in line 91 of the manuscript, I think you meant equations 1 and 2. Please confirm.

Experimental design

The experimental design appears rigorous and detailed. However, the description of the dataset in Section 4, Lines 339-343 can be improved. It is not ideal to expect to readers to refer to other sources for further detail on the dataset used in the study.

Validity of the findings

Please include the AUC similar to the PRC, in a graphical format. This will provide a balanced evaluation of the performance.

Additional comments

Please consider revising the study from a forensic perspective to a generic perspective, as the current argument has a very little forensic presentation.

---

## Round 0.2 · accepted · Accept

Dear Authors,

Two of the reviewers of previous round did not respond to the invitation for the last revision. You seem to have addressed the reviewers' comments. I think your manuscript is ready for publication.

Best wishes,

·

Basic reporting

Clear and unambiguous, professional English used throughout.

Experimental design

Original primary research within Aims and Scope of the journal.

Validity of the findings

Impact and novelty not assessed. Meaningful replication encouraged where rationale & benefit to literature is clearly stated.

Cite this review as

Reviewer 2 ·

Basic reporting

Concept is clear and related to journal

Experimental design

Methodology used in correct

Validity of the findings

Conclusion is well stated